# The Emerging Importance of Cirsimaritin in Type 2 Diabetes Treatment

**DOI:** 10.3390/ijms24065749

**Published:** 2023-03-17

**Authors:** Abdelrahim Alqudah, Rabaa Y. Athamneh, Esam Qnais, Omar Gammoh, Muna Oqal, Rawan AbuDalo, Hanan Abu Alshaikh, Nabil AL-Hashimi, Mohammad Alqudah

**Affiliations:** 1Department of Clinical Pharmacy and Pharmacy Practice, Faculty of Pharmaceutical Sciences, The Hashemite University, Zarqa 13133, Jordan; 2Department of Medical Laboratory Sciences, Faculty of Allied Science, Zarqa University, Zarqa 13110, Jordan; 3Department of Biology and Biotechnology, Faculty of Science, The Hashemite University, Zarqa 13133, Jordan; 4Department of Clinical Pharmacy and Pharmacy Practice, Faculty of Pharmacy, Yarmouk University, Irbid 21163, Jordan; 5Department of Pharmaceutical Technology, Faculty of Pharmaceutical Sciences, The Hashemite University, Zarqa 13133, Jordan; 6Prince Hamza Hospital, Amman 11123, Jordan; 7Department of Pharmaceutical Chemistry, Faculty of Pharmaceutical Sciences, The Hashemite University, Zarqa 13133, Jordan; 8Physiology Department, School of Medicine and Biomedical Sciences, Arabian Gulf University, Manama 26671, Bahrain

**Keywords:** cirsimaritin, insulin resistance, type 2 diabetes, oxidative stress, inflammation

## Abstract

Cirsimaritin is a dimethoxy flavon that has different biological activities such as antiproliferative, antimicrobial, and antioxidant activities. This study aims to investigate the anti-diabetic effects of cirsimaritin in a high-fat diet and streptozotocin-(HFD/STZ)-induced rat model of type 2 diabetes mellitus (T2D). Rats were fed HFD, followed by a single low dose of STZ (40 mg/kg). HFD/STZ diabetic rats were treated orally with cirsimaritin (50 mg/kg) or metformin (200 mg/kg) for 10 days before terminating the experiment and collecting plasma, soleus muscle, adipose tissue, and liver for further downstream analysis. Cirsimaritin reduced the elevated levels of serum glucose in diabetic rats compared to the vehicle control group (*p <* 0.001). Cirsimaritin abrogated the increase in serum insulin in the treated diabetic group compared to the vehicle control rats (*p <* 0.01). The homeostasis model assessment of insulin resistance (HOMA-IR) was decreased in the diabetic rats treated with cirsimaritin compared to the vehicle controls. The skeletal muscle and adipose tissue protein contents of GLUT4 (*p* < 0.01 and *p* < 0.05, respectively) and pAMPK-α1 (*p* < 0.05) were upregulated following treatment with cirsimaritin. Cirsimaritin was able to upregulate GLUT2 and AMPK protein expression in the liver (*p* < 0.01, <0.05, respectively). LDL, triglyceride, and cholesterol were reduced in diabetic rats treated with cirsimaritin compared to the vehicle controls (*p* < 0.001). Cirsimaritin reduced MDA, and IL-6 levels (*p* < 0.001), increased GSH levels (*p* < 0.001), and reduced GSSG levels (*p* < 0.001) in diabetic rats compared to the vehicle control. Cirsimaritin could represent a promising therapeutic agent to treat T2D.

## 1. Introduction

Diabetes is a multifactorial metabolic syndrome characterized by abnormalities in carbohydrates, fat, and protein metabolisms, which lead to hyperglycemia [1]. The most common type of diabetes mellitus is type 2 diabetes (T2D) which is characterized by impaired glucose utilization by skeletal muscle, liver, and adipose tissue resulting from glucose intolerance due to insulin resistance accompanied by insulin deficiency as a result of islet beta-cells injury [2]. Genetic disposition, environmental factors, diet, physical inactivity, and obesity are risk factors that contribute significantly to the progression of insulin resistance and T2D development [3,4,5].

Inflammation and oxidative stress are important biological factors in the pathogenesis of T2D development and its complications [6]. It is not well understood how inflammation contributes to the pathogenesis of T2D; however, it has been observed that pro-inflammatory cytokines such as interleukin-6 (IL-6) are synthesized by adipose tissue, which increases as fat body mass increases, which leads to the development of insulin resistance [7]. Moreover, a large body of evidence has proven that oxidative stress plays a key role in the etiology of T2D [1,8]. The chronic exposure of cells and tissue to hyperglycemia results in the non-enzymatic glycation of proteins, which leads to the production of reactive oxygen species which may cause DNA damage [9]. Furthermore, oxidative stress was found to inhibit the promoter activity and mRNA expression of the insulin gene in pancreatic islet cells, leading to a decrease in insulin gene expression [10]. In addition, oxidative stress is also involved in inducing insulin resistance, thus increasing the incidence of T2D development [11]. Taken together, T2D etiology and its complications are strongly linked to inflammation and oxidative stress.

Glucose disposal is mediated by the effect of insulin on its receptors on the cell surfaces of insulin-sensitive tissues, particularly skeletal muscle, adipose tissues, and the liver [12,13]. Once the insulin receptor is activated, glucose transport over the plasma membrane occurs through different members of the glucose transporter (GLUT) family, such as GLUT 4, which is expressed in skeletal muscle and adipose tissue, and GLUT2, which is expressed in the liver [14]. Insulin resistance develops when insulin signaling is impaired; however, insulin resistance is compensated initially by increasing insulin secretion, but eventually, insulin release from pancreatic β-cells becomes insufficient for maintaining normal blood glucose concentration, which leads to T2D [15]. Several studies have shown that the activation of AMP-activated protein kinase (AMPK) stimulates the translocation of GLUT4 and GLUT2 to the cell surface, which increases glucose uptake through an insulin-independent pathway [16,17,18]. Thus, enhancing insulin sensitivity could be achieved through the activation of the AMPK signaling pathway.

Cirsimaritin is a dimethoxy flavone (Figure 1) that is found in different plants such as *Lithocarpus dealbatus, Artemisia Judaica, Microtea debilis, Cirsium japonicum,* and *Ocimum sanctum* [19]. Previous reports show that cirsimaritin has different biological activities such as antimicrobial, antispasmodic, and antiproliferative activities [20,21]. Moreover, extracts from rosemary leaves containing cirsimaritin showed high antioxidant activity [22,23]. Additionally, cirsimaritin also showed anti-inflammatory activity by inhibiting nitric oxide production and the inducible expression of nitric oxide synthase, in addition to blocking different cytokines, including IL-6 and tumor necrosis factor-α (TNF-α) [24]. Furthermore, several studies showed that cirsimaritin has an anti-diabetic effect. One study demonstrated that extracts from *T. polium* containing cirsimaritin had an insulinotropic effect on a rat insulinoma cell line, INS1E cells. Moreover, the extracts containing cirsimaritin were able to reduce glucose levels significantly in hyperglycemic rats [25]. Another study performed with visual screening revealed that cirsimaritin has a good affinity for blocking dipeptidyl peptidase 4 (DDP-4), which will increase insulin secretion [26]. Additionally, cirsimaritin showed an ability to enhance glucose uptake rate in TNF-α-treated mouse FL83B hepatocytes, suggesting that cirsimaritin might improve insulin resistance in the liver [27]. These results show that cirsimaritin has an antidiabetic effect, but more research is needed to determine the applicability of cirsimaritin in diabetes mellitus treatment. Therefore, this study aims to assess the effect of cirsimaritin on AMPK-GLUT4 and AMPK-GLUT2 pathways in a high-fat diet (HFD)/STZ-induced rat model.

## 2. Results

### 2.1. The Hypoglycemic Effect of Cirsimaritin

Serum glucose was significantly higher in the vehicle control diabetic group compared to the non-diabetic group (Figure 2A, n = 6, *p* < 0.001). Treatment with cirsimaritin significantly reduced serum glucose concentrations compared to the vehicle controls in the presence of T2D (Figure 2A, n = 6, *p* < 0.001). Similarly, the mean glucose concentration in the diabetic group was significantly reduced with metformin treatment compared to the vehicle controls (Figure 2A, n = 6, *p* < 0.01). However, glucose concentration was significantly reduced with metformin treatment compared to cirsimaritin treatment (Figure 2A, n = 6, *p* < 0.05). Insulin levels were significantly increased in the vehicle control diabetic group compared to the non-diabetic group (Figure 2B, n = 6, *p* < 0.001); however, treating diabetic rats with cirsimaritin significantly reduced the insulin compared to the vehicle control group (Figure 2B, n = 6, *p* < 0.01). Like cirsimaritin, treating diabetic rats with metformin significantly reduced insulin levels compared to the vehicle control group (Figure 2B, n = 6, *p* < 0.001). However, metformin significantly reduced insulin levels compared to the cirsimaritin-treated group (Figure 2B, n = 6, *p* < 0.05).

To determine the effects of cirsimaritin on insulin resistance, HOMA-IR was measured. The presence of T2D was confirmed via HOMA-IR, which significantly increased in the vehicle control diabetic group compared to the non-diabetic group (Figure 2C, n = 6, *p* < 0.001). Interestingly, cirsimaritin was able to restore HOMA-IR in the diabetic group, which was comparable to the non-diabetic group. The same effect on HOMA-IR was observed when diabetic rats were treated with metformin. HOMA-IR was significantly reduced with metformin treatment compared to cirsimaritin treatment (Figure 2C, n = 6, *p* < 0.01). Moreover, the blood glucose level during IPGTT was significantly lower in the cirsimaritin and metformin groups compared to the vehicle control (Figure 3, n = 6, *p* < 0.001).

To determine the mechanism by which cirsimaritin improves blood glucose and insulin resistance, GLUT4 and phosphorylated AMPK (pAMPK-α1) protein expression were measured in skeletal muscle tissue. GLUT4 protein expression was significantly downregulated in the presence of T2D (Figure 4A, n = 6, *p* < 0.001); however, treating diabetic rats with cirsimaritin significantly upregulated GLUT4 expression compared to the vehicle control diabetic group (Figure 4A, n = 6, *p* < 0.01). Metformin was also able to upregulate GLUT4 expression compared to the vehicle control diabetic group (Figure 4A, n = 6, *p* < 0.001). Metformin treatment was able to upregulate GLUT4 expression significantly higher compared to cirsimaritin treatment (Figure 4A, n = 6, *p* < 0.05). Similarly, pAMPK-α1 expression was significantly downregulated as a result of T2D (Figure 4B, n = 6, *p* < 0.001), which was abrogated with cirsimaritin or metformin (Figure 4B, n = 6, *p* < 0.05, < 0.001, respectively). pAMPK-α1 was significantly higher with metformin treatment compared to cirsimaritin treatment (Figure 4B, n = 6, *p* < 0.05).

Moreover, GLUT4 and pAMPK-α1 protein expression in adipose tissue were measured. GLUT4 protein expression was significantly downregulated in the presence of T2D (Figure 5A, n = 6, *p* < 0.001); whereas the administration of cirsimaritin significantly upregulated GLUT4 expression compared to the vehicle controls (Figure 5A, n = 6, *p* < 0.05). Metformin was also able to upregulate GLUT4 expression compared to the vehicle control diabetic group (Figure 5A, n = 6, *p* < 0.01). GLUT4 expression in adipose tissue was significantly upregulated with the metformin treatment compared to the cirsimaritin treatment (Figure 5A, n = 6, *p* < 0.001). Similarly, pAMPK-α1 expression was significantly downregulated as a result of T2D (Figure 5B, n = 6, *p* < 0.001), and treating diabetic rats with either cirsimaritin or metformin significantly upregulated pAMPK-α1 expression compared to the vehicle control diabetic group (Figure 5B, n = 6, *p* < 0.05, < 0.001, respectively). No difference was observed in pAMPK-α1 expression between the cirsimaritin and metformin groups.

To further elucidate the mechanism by which cirsimaritin improves insulin sensitivity and reduces glucose levels, hepatic glucose transporter 2 (GLUT2) and pAMPK-α1 protein expression were measured. GLUT2 protein expression was significantly downregulated in the presence of T2D (Figure 6A, n = 6, *p* < 0.001), whereas cirsimaritin significantly upregulated GLUT2 expression compared to the vehicle control diabetic group (Figure 6A, n = 6, *p* < 0.01). Metformin was also able to upregulate GLUT2 expression compared to the vehicle controls (Figure 6A, n = 6, *p* < 0.001). Similarly, pAMPK-α1 expression was significantly downregulated as a result of T2D (Figure 6B, n = 6, *p* < 0.001), and treating diabetic rats with either cirsimaritin or metformin significantly upregulated pAMPK-α1 expression compared to the vehicle controls (Figure 6B, n = 6, *p* < 0.05, < 0.001, respectively). No difference was observed in GLUT2 and pAMPK-α1 expression between the cirsimaritin and metformin groups.

### 2.2. The Effect of Cirsimaritin on the Lipid Profile

As depicted in Figure 7, dyslipidemia was present in diabetic rats. LDL (Figure 7A, n = 6, *p* < 0.001), total cholesterol (Figure 7B, n = 6, *p* < 0.05), and triglycerides (TGs; Figure 7C, n = 6, *p* < 0.001) were significantly higher in the vehicle control diabetic group, compared to the non-diabetic group. Treating diabetic rats with cirsimaritin significantly reduced serum LDL (Figure 7A, n = 6, *p* < 0.001), total cholesterol (Figure 7B, n = 6, *p* < 0.001), and TGs (Figure 7C, n = 6, *p* < 0.001) systemic concentration, compared to the vehicle control diabetic group. Similarly, metformin was able to significantly reduce LDL (Figure 7A, n = 6, *p* < 0.001), cholesterol (Figure 7B, n = 6, *p* < 0.01), and TGs (Figure 7C, n = 6, *p* < 0.001) levels in diabetic rats, compared to the vehicle controls. Interestingly, LDL and TG systemic concentrations were significantly lower in the cirsimaritin group compared to the metformin group (Figure 7A,C, n = 6, *p* < 0.05, <0.001, respectively).

### 2.3. The Effects of Cirsimaritin on GSH, GSSG, MDA, and IL-6

Serum GSH expression was significantly reduced in vehicle-control diabetic rats, compared to non-diabetic rats (Figure 8A, n = 6, *p* < 0.001); however, treating diabetic rats with either cirsimaritin or metformin demonstrated a significant increase in GSH compared to the vehicle control diabetic group (Figure 8A, n = 6, *p* < 0.001, < 0.01, respectively). Moreover, serum GSSG was significantly increased in vehicle-control diabetic rats, compared to non-diabetic rats (Figure 8B, n = 6, *p* < 0.001); however, cirsimaritin and metformin were able to reduce GSSG levels in diabetic rats, compared to the vehicle control (Figure 8B, n = 6, *p* < 0.001). On the other hand, serum MDA (Figure 8C, n = 6, *p* < 0.001) and IL-6 (Figure 8D, n = 6, *p* < 0.001) concentrations were significantly increased in the presence of T2D. Interestingly, cirsimaritin showed an ability to reduce serum MDA and IL-6 levels significantly in diabetic rats, in comparison to the vehicle control group (Figure 8C,D, n = 6, *p* < 0.001). The same effect was observed when diabetic rats were treated with metformin (Figure 8C,D, n = 6, *p* < 0.001). The IL-6 systemic level was significantly lower in the metformin group compared to the cirsimaritin group (Figure 8D, n = 6, *p* < 0.05).

## 3. Discussion

Food value is linked to nutritional contents and digestibility, as well as the presence or absence of toxic ingredients [28]. Indeed, the bionutrients that foods offer are essential for life, but also, foods possess other bioactive compounds that are important for health promotion and disease prevention [29]. The consumption of a healthy diet is strongly correlated with a reduction in the risk of several chronic diseases such as cancer, diabetes, cardiovascular diseases and atherosclerosis, neurodegenerative disorders, and inflammation, as well as their complications [30]. These therapeutic properties of food gave rise to medicinal drugs made from certain kinds of food, particularly plants. Numerous medicinal plants have been largely used in the treatment of several diseases, such as cardiovascular diseases [31], cancer [32], diabetes [33], inflammation [34], depression [35,36], and pain management [37]. Therefore, the aim of this study was to assess, for the first time, the antidiabetic efficacy of flavonoid compound, cirsimaritin.

The findings of our study revealed that cirsimaritin significantly reduced glucose, insulin, and HOMA-IR in an HFD/STZ-induced diabetic rat model. In addition, cirsimaritin improved insulin resistance by upregulating GLUT4 and AMPK expression in soleus muscle and adipose tissue. Furthermore, cirsimaritin upregulated GLUT2 and AMPK expression in the liver. Moreover, cirsimaritin exerts antioxidant and anti-inflammatory effects.

Hyperglycemia and insulin resistance are hallmarks of T2D, and they are implicated in T2D complications such as nephropathy, neuropathy, and retinopathy [38]. Our findings demonstrated that cirsimaritin reduced glucose, insulin, and HOMA-IR. This finding is consistent with the previous literature [39] where cirsimaritin enhanced the glucose uptake rate in diabetic mice hepatocytes.

Skeletal muscles and adipose tissue consume a high portion of blood glucose. Several chronic conditions and mechanisms underlie insulin resistance (IR) in these tissues [40,41]. Many studies have linked insulin resistance to impaired GLUT4-mediated glucose uptake [42,43,44]. According to animal models, the reduction in GLUT4 expression was 50% in the skeletal muscles of the hypertriglyceridemia insulin resistance mouse model [43]. Moreover, a 70% reduction in GLUT4 protein expression in the adipose-specific genetic knockout mouse model was associated with insulin resistance [45]. The activation of the AMPK-GLUT4 pathway enhances insulin sensitivity, and it has been shown to improve glucose control in T2D [46,47]. Moreover, GLUT2 is responsible for glucose uptake in the liver, and it is required for the physiological control of glucose-sensitive genes. The inactivation of GLUT2 in the liver leads to impaired glucose-stimulated insulin secretion [13,48]. The findings of the present study demonstrated that cirsimaritin-improved insulin resistance is mediated by the activation of the AMPK-GLUT4 pathway in the skeletal and adipose tissues, and by the activation of the AMPK-GLUT2 in the liver.

Dyslipidemia is tightly associated with T2D, and it is a major risk factor leading to T2D-associated complications [49]. Several studies have linked dyslipidemia to microvascular complications associated with T2D, such as diabetic retinopathy, nephropathy, and neuropathy [50]. The present study showed that cirsimaritin improved the lipid profile in high-fat diabetic rats. Interestingly, cirsimaritin-treated animals demonstrated lower LDL-C and TG levels after 10 days of treatment, compared to the vehicle control group. To our knowledge, this is the first study that shows the lipid-lowering effects of pure cirsimaritin. Previous studies using extracted flavonoids, including cirsimaritin, failed to demonstrate similar findings [25]. Although the exact mechanism is yet to be unrevealed, however, these findings highlights could pave the way to further investigate cirsimaritin in T2D and dyslipidemia.

Oxidative stress is implicated in the early stages of T2D [51]. Oxidative stress is referred to as an overproduction of reactive oxygen species (ROS), and a reduction in the rate of antioxidant defense mechanisms such as GSH (a non-enzymatic antioxidant) [52]. Additionally, ROS induces the release of MDA, a highly reactive compound that interacts with proteins and nucleic acids, and that causes damage to various tissues and cells [53]. MDA has been used as a biomarker of lipid peroxidation and as an indication of free radical damage in the blood [54]. Our findings indicate that similar to metformin, cirsimaritin was able to decrease elevated MDA levels in the high-fat T2D rat model. Furthermore, cirsimaritin was able to restore the GSH/GSSG balance that was disturbed in the high-fat T2D rat model. Taken together, these findings suggest a powerful antioxidant effect of cirsimaritin. Numerous studies have pointed out the antioxidant effects of cirsimaritin in different assays, as revised in [55]. In those studies, cirsimaritin extracted from *Teucrium ramosissimum* showed an excellent antioxidant activity, using ABTS assay with a Teac value of 2.04 μM. In addition, cirsimaritin extracted from *Cirsium japonicum* inhibited DPPH free radicals, with a percentage of between 80% to 100% at a dose of 100 μg/mL. Moreover, cirsimaritin extracted from *Combretum fragrans* showed potent DPPH radicals scavenging activity. It is postulated that as a flavonoid, cirsimaritin has a broad range of biological actions that could be related to the direct scavenging of ROS [55].

Subclinical chronic inflammation has been implicated in the development and progression of insulin resistance, T2D, and its complications [56]. In particular, the increased levels of the multifunctional cytokine, IL-6, have been linked to the pathogenesis of T2D. Our findings demonstrate that cirsimaritin reduced the circulating IL-6 levels with respect to the untreated group. Previous evidence has shown that cirsimaritin inhibited the production of several cytokines such as IL-6 and TNF-α via transcriptional factor-mediated mechanisms that downregulate gene expression [24].

The limitations of this study include the following aspects: (i) cirsimaritin was administered for a short period of time, and (ii) GLUT4 expression was assessed using immunoblotting that was reflective of its total amount; however, immunohistochemistry may be a better technique for assessing its activity and translocation to the cell membrane. Nevertheless, our findings in this study indicate the important role of cirsimaritin in improving the typical features of T2D.

## 4. Materials and Methods

### 4.1. The Induction of T2D and Experimental Design

Animal experimental procedures were approved by the animal ethics committee at the Hashemite University (IRB number: 14/4/2021/2022, 14 April 2022), and were in accordance with the guidelines of the US National Institutes of Health on the use and care of laboratory animals, and with the Animal Research: Reporting of in Vivo Experiments (ARRIVE) guidelines (https://arriveguid elines.org, accessed on 2 May 2022).

Thirty-week-old male adult Sprague-Dawley rats (average weight 264 ± 3.5) were maintained under standard conditions, including 12 h light/dark cycles and a 22 ± 2° temperature [57]. T2D was induced by feeding the experimental rats HFD (60% fat) for 3 weeks, followed by one intraperitoneal injection of streptozotocin (STZ; 40 mg/kg) [58]. The average weight after diabetes induction for the mice fed with HFD was 328.4 ± 4.4. One week after STZ injection, plasma glucose was measured, and rats with a plasma glucose concentration over 200 mg/dL were considered to have developed T2D, and were selected for the subsequent experiments.

Rats were randomly divided into four groups (n = 6 each), as follows: (i) normal non-diabetic control group (non-diabetic, ND) receiving normal diet, (ii) vehicle control (VC) diabetic group treated with dimethyl sulfoxide (DMSO, Panreac Quimica SA, Barcelona, Spain) only, (iii) diabetic group treated with 50 mg/kg cirsimaritin (Sigma-Aldrich, Dorset, UK), and (iv) diabetic group treated with 200 mg/kg metformin (MeRCK, Darmstadt, Germany); the dose of metformin was in line with a previous study that used 200 mg/kg metformin for the treatment of diabetic rats [59]. The dose of cirsimaritin used in this study was based on a previous report showing that the doses between 50–200 mg/kg did not produce any noticeable side effects; thus, the lowest dose (50 mg/kg) was chosen for this study [60]. All treatments were given orally once per day. After 10 days of treatment, rats were euthanized as per local standard operating procedures, using carbon dioxide before blood and skeletal muscle (soleus muscle), adipose tissue, and liver were collected for ex-vivo analysis.

### 4.2. Biochemical Investigations

#### 4.2.1. Measurements of Serum Glucose, Insulin, and Lipids in Rat Serum

Serum glucose and insulin were determined using a commercial kits glucose assay kit (Mybiosource, San Diego, CA, USA), and a rat insulin ELISA kit (Mybiosource, San Diego, CA, USA), respectively, as per the manufacturer’s instructions. Triglyceride (TG, triglyceride assay kit), low-density lipoproteins (LDL, LDL assay kit), and cholesterol (Total Cholesterol assay kit) were also measured using commercially available kits (MybioSource, San Diego, CA, USA) according to the manufacturer’s instructions.

#### 4.2.2. Homeostasis Model Assessment of Insulin Resistance (HOMA-IR)

This model represents the interaction between fasting plasma insulin and fasting plasma glucose, which is a useful tool for determining insulin resistance. In the current study, we used the following formula to compute HOMA-IR [61]:HOMA-IR=(Fasting glucose (mg/dL)×Fasting insulin (μIU/mL)/405

#### 4.2.3. Intraperitoneal Glucose Tolerance Test

Rats were given an intraperitoneal injection of glucose (0.5 g/kg) after being fasted for 18 h. Using a glucometer, blood glucose levels were measured from the tail vein at 0, 30, 60, and 120 min (Accu-Check Performa, Roche Diagnostics).

#### 4.2.4. Measurement of Serum Glutathione (GSH), Oxidized Glutathione (GSSG), Malondialdehyde (MDA), and IL-6 Serum Concentrations

Reduced glutathione (GSH, GSH assay kit), oxidized glutathione (GSSG, GSSG assay kit), and IL-6 (IL-6 ELISA kit) levels were measured in the serum using commercially available kits (Mybiosource, San Diego, CA, USA). Plasma MDA level was determined by using a commercially available thiobarbituric acid (TBA) Assay Kit (Mybiosource, San Diego, CA, USA) according to the manufacturer’s instructions.

### 4.3. Western Blotting

Skeletal muscle tissues (soleus muscle), adipose tissue, and liver were homogenized in radioimmunoprecipitation (RIPA)-lysis buffer, containing a protease inhibitor cocktail (Santa Cruz Biotechnology, USA), using a tissue homogenizer. Homogenates were centrifuged at 13,000 rpm for 20 min at 4 °C, and the supernatant was collected. Total protein was quantified using a bicinchoninic acid assay (Bioquochem, Austurias, Spain). Equal amounts of protein were separated using a sodium dodecyl sulfate–polyacrylamide gel, then transferred onto a nitrocellulose membrane (Thermo Fisher Scientific, Carlsbad, CA, USA). The membrane was blocked for 1 h at room temperature using 3% bovine serum albumin (BSA), before incubating overnight with either pAMPK-α1 (Abcam, Cambridge, UK), GLUT2, or GLUT4 (Mybiosource, San Diego, CA, USA) primary antibodies (1:1000 dilution). The membrane was washed three time with a washing buffer (Tween-20/Tris-buffered saline) before incubating it with the goat-anti-rabbit secondary antibody (Mybiosource, San Diego, CA, USA, 1:5000 dilution) for 1 h at room temperature. Following incubation, the membrane was washed three times before submerging it into the ECL substrate (ThermoScientific, Carlsbad, CA, USA) for one minute, followed by imaging with the chemiLITE Chemiluminescence Imaging System (cleaverscientific, Rugby, UK). To ensure equal protein gel loading, β-actin was used as a housekeeping gene (Mybiosource, San Diego, CA, USA, 1:10,000 dilution). The intensities of the bands were measured using Image J software, and adjusted to β-actin.

### 4.4. Statistical Analysis

All analyzed parameters were tested for the normality of the data using the Kolmogorov-Smirnov test. Data are represented as mean ± SEM. Differences between groups were calculated using a one-way analysis of variance (ANOVA) or two-way ANOVA, followed by a Tukey post hoc test using PGraphPad Prism software version (9.3.1). The significance value of difference was considered when the *p*-value was less than 0.05.

## 5. Conclusions

Our findings in this study revealed that cirsimaritin might be a very useful agent for the treatment of T2D due to its ability to reduce insulin resistance, and its activation of the GLUT4-AMPK and GLUT2-AMPK pathways in skeletal muscle, adipose tissue, and liver. In addition, cirsimaritin could be a useful agent for improving lipid profiles and for reducing oxidative stress. Cirsimaritin effects and mechanisms were like metformin, the gold standard for T2D treatment.

## Figures and Tables

**Figure 1 ijms-24-05749-f001:**
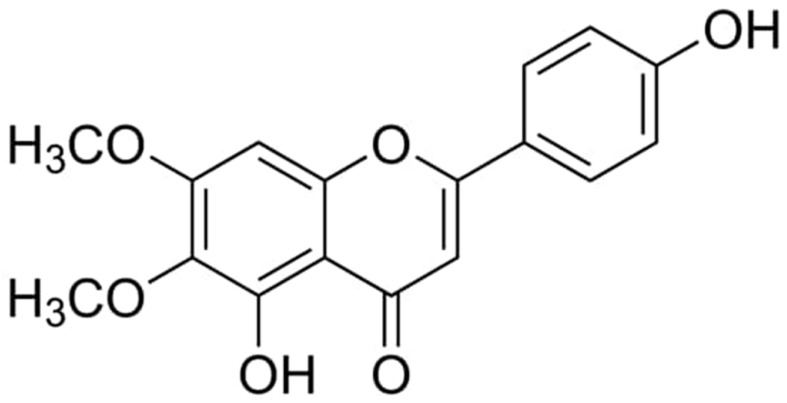
Chemical structure of cirsimaritin.

**Figure 2 ijms-24-05749-f002:**
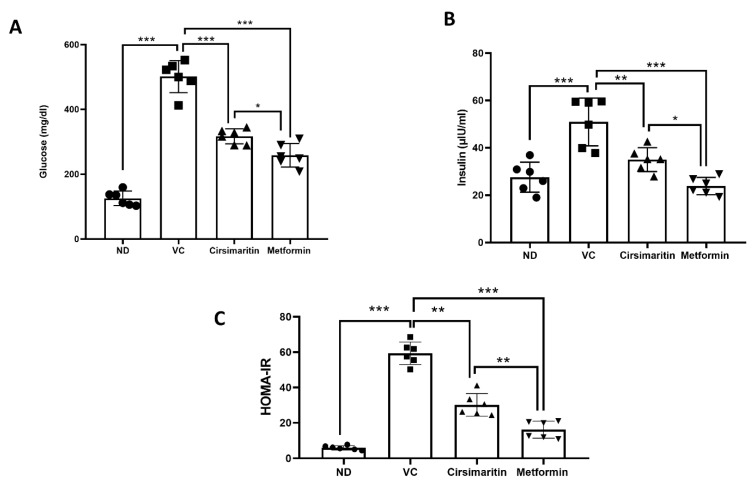
The anti-diabetic effect of cirsimaritin. Cirsimaritin significantly reduced glucose (**A**) and insulin (**B**) levels in diabetic rats. HOMA-IR (**C**) was significantly reduced with cirsimaritin treatment. Rats were fed HFD for 3 weeks, followed by a single dose of STZ injection (40 mg/kg); and once diabetes was confirmed, rats were treated with 50 mg/kg cirsimaritin or 200 mg/kg metformin for 10 days. After the end of the experiment, serum was collected for ELISA analyses. One-way ANOVA was followed by Tukey post hoc multiple comparison test, * *p* < 0.05, ** *p* < 0.01, *** *p* < 0.001. ND; non-diabetic, VC; vehicle control.

**Figure 3 ijms-24-05749-f003:**
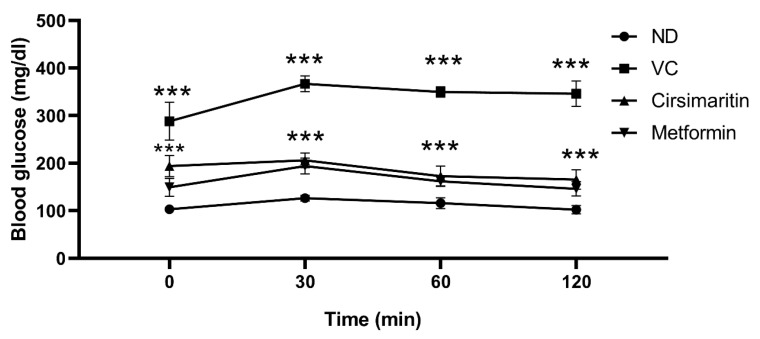
Cirsimaritin reduced glucose levels during the intraperitoneal glucose tolerance test (IPGTT). Rats were fed with HFD for 3 weeks, followed by a single dose of STZ injection (40 mg/kg); once diabetes was confirmed, rats were treated with 50 mg/kg cirsimaritin or 200 mg/kg metformin for 10 days. Rats then fasted overnight before injection with 0.5 g/kg glucose intraperitoneally, and glucose levels were determined at 0, 30, 60, and 120 min. Two-way ANOVA followed by Tukey post hoc multiple comparison test, *** *p* < 0.001. ND; non-diabetic, VC; vehicle control.

**Figure 4 ijms-24-05749-f004:**
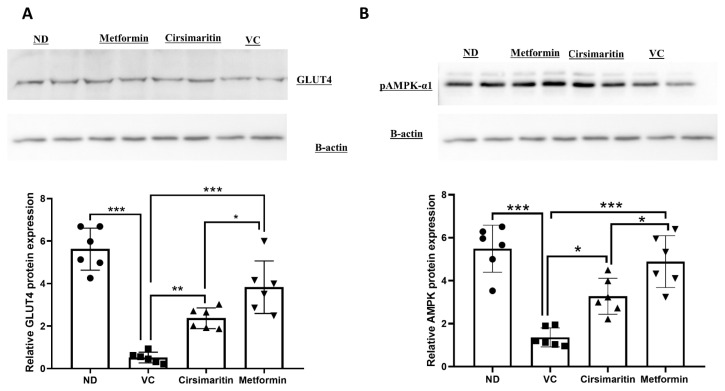
Cirsimaritin upregulated GLUT4 and pAMPK-α1 expression in the soleus muscle. Cirsimaritin significantly upregulated GLUT4 (**A**) and pAMPK-α1 (**B**) expression within soleus muscle in diabetic rats. Rats were fed HFD for 3 weeks, followed by a single dose of STZ injection (40 mg/kg); once diabetes was confirmed, rats were treated with 50 mg/kg cirsimaritin or 200 mg/kg metformin for 10 days. Rats were then sacrificed and after the end of the experiment following euthanasia, soleus muscle was isolated and homogenized for downstream Western blotting. One-way ANOVA was followed by Tukey post hoc multiple comparison test, * *p* < 0.05, ** *p* < 0.01, *** *p* < 0.001. ND; non-diabetic, VC; vehicle control.

**Figure 5 ijms-24-05749-f005:**
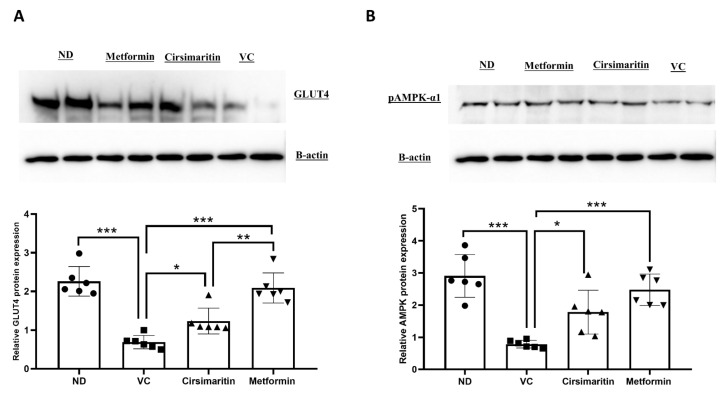
Cirsimaritin upregulated GLUT4 and pAMPK-α1 expression in adipose tissue. Cirsimaritin significantly upregulated adipose tissue GLUT4 (**A**) and pAMPK-α1 (**B**) expression in diabetic rats. Rats were fed HFD for 3 weeks, followed by a single dose of STZ injection (40 mg/kg); after diabetes was confirmed, rats were treated with 50 mg/kg cirsimaritin or 200 mg/kg metformin for 10 days. At the end of the experiment following euthanasia, adipose tissue was isolated and homogenized before Western blotting was performed. One-way ANOVA was followed by Tukey post hoc multiple comparison test, * *p* < 0.05, ** *p* < 0.01, *** *p* < 0.001. ND; non-diabetic, VC; vehicle control.

**Figure 6 ijms-24-05749-f006:**
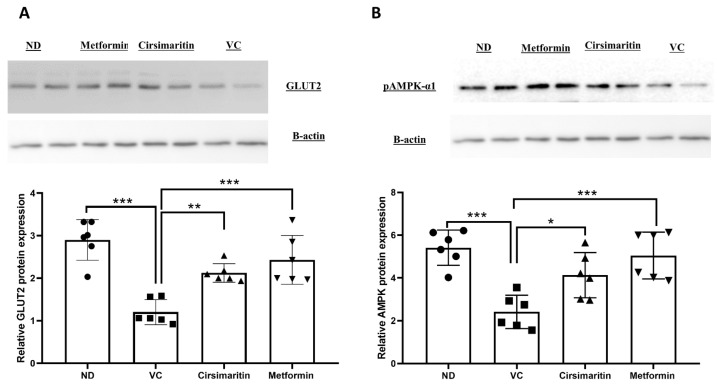
Cirsimaritin upregulated GLUT2 and pAMPK-α1 expression in the liver. Cirsimaritin significantly upregulated hepatic GLUT2 (**A**) and pAMPK-α1 (**B**) expression in diabetic rats. Rats were fed HFD for 3 weeks, followed by a single dose of STZ injection (40 mg/kg); and once diabetes was confirmed, rats were treated with 50 mg/kg cirsimaritin or 200 mg/kg metformin for 10 days. At the end of the experiment following euthanasia, the liver was isolated and homogenized before Western blotting performed. One-way ANOVA was followed by Tukey post hoc multiple comparison test, * *p* < 0.05, ** *p* < 0.01, *** *p* < 0.001. ND; non-diabetic, VC; vehicle control.

**Figure 7 ijms-24-05749-f007:**
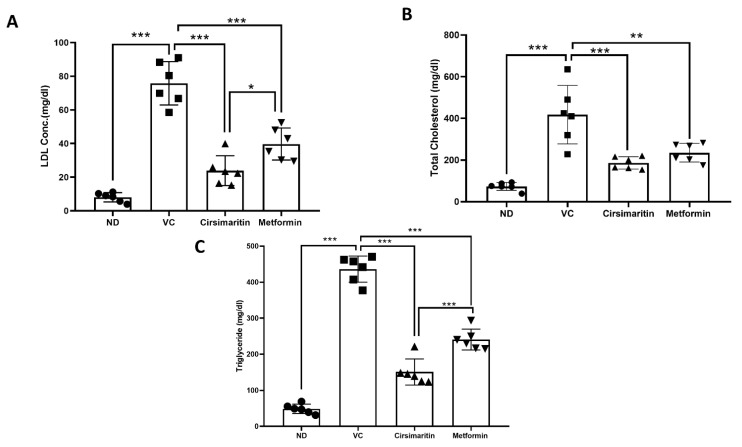
Cirsimaritin improves lipid profile in diabetes. Cirsimaritin significantly reduced LDL (**A**), total cholesterol (**B**), and triglyceride (**C**) systemic concentrations in diabetic rats. Rats were fed HFD for 3 weeks, followed by a single dose of STZ injection (40 mg/kg), and once diabetes was confirmed, rats were treated with 50 mg/kg cirsimaritin or 200 mg/kg metformin for 10 days. At the end of the experiment following euthanasia, serum was collected for ELISA analyses. One-way ANOVA was followed by Tukey post hoc multiple comparison test, * *p* < 0.05, ** *p* < 0.01, *** *p* < 0.001. ND; non-diabetic, VC; vehicle control.

**Figure 8 ijms-24-05749-f008:**
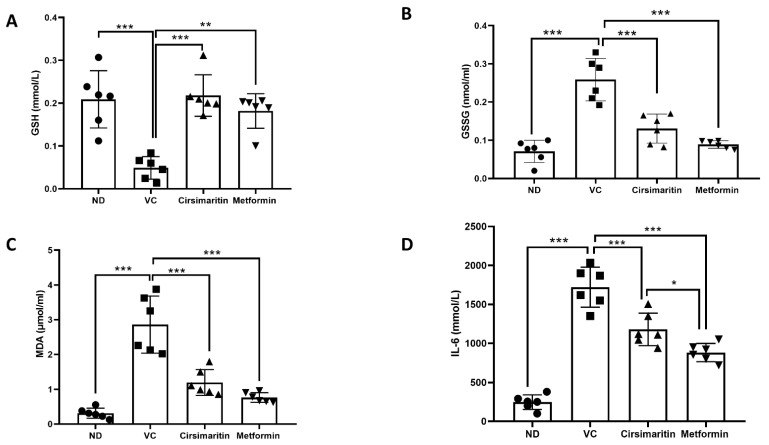
The antioxidant and anti-inflammatory effect of cirsimaritin. Cirsimaritin significantly increased GSH (**A**) and reduced GSSG (**B**), and reduced MDA (**C**) and IL-6 (**D**) systemic concentrations in diabetic rats. Rats were fed HFD for 3 weeks, followed by a single dose of STZ injection (40 mg/kg), and once diabetes was confirmed, rats were treated with 50 mg/kg cirsimaritin or 200 mg/kg metformin for 10 days. After the end of the experiment following euthanasia, serum was collected for ELISA analysis. One-way ANOVA was followed by Tukey post hoc multiple comparison test, * *p* < 0.05, ** *p* < 0.01, *** *p* < 0.001. ND; non-diabetic, VC; vehicle control.

## Data Availability

The datasets generated during and/or analyzed during the current study are available from the corresponding author upon reasonable request.

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
