# Peer review of "The Emerging Importance of Cirsimaritin in Type 2 Diabetes Treatment"

_ijms, 2023, doi:10.3390/ijms24065749_

Round 1
Reviewer 1 Report
I have no additional comment
Author Response
Dear editor and reviewers
We would like to thank you for the invested time and effort in carefully reviewing our manuscript.
Reviewer 2 Report
Manuscript ID: ijms-2255817
Title: The emerging importance of cirsimaritin in type 2 diabetes treatment
I have reviewed the manuscript entitled “The emerging importance of cirsimaritin in type 2 diabetes treatment”, and it shows an interesting approach on the use of cirsimaritin in the treatment of type 2 diabetes in rats. The methodology employed is adequate, the idea of the manuscript is original and the subject is suitable for the International Journal of Molecular Sciences. However, some grammar and proofing issues need to be considered before possible publication in the IJMS.
Abstract, line 5: It is unclear whether two dosages of cirsimaritin were evaluated.
Introduction: Species names in the last paragraph of the Introduction must be italicized.
Results: There are captions for Figures 5 and 7 but the images were not provided in the manuscript.
Discussion: The discussion of the manuscript needs to be improved. The authors can better explain the importance of new medicines based on natural products. You can read on that in: https://doi.org/10.1007/s11011-017-0089-y and https://doi.org/10.1371/journal.pone.0255996
Material and methods: Can you provide the average weight of the rats before and after the T2D induction?
Material and methods: It is necessary to explain how cirsimaritin was obtained and purified. Was it purchased from a commercial source (insert manufacturer) or extracted from a plant (insert methodology employed)?
Manuscript grammar must be carefully proofread.
Author Response
Dear editor and reviewers
We would like to thank you for the invested time and effort in carefully reviewing our manuscript. We are grateful for giving us the opportunity to revise our manuscript. Your comments were very useful and helped us in improving our manuscript. After careful consideration of the comments, the revision included many positive changes as suggested. Changes are highlighted using track changes.
Title: The emerging importance of cirsimaritin in type 2 diabetes treatment
C: I have reviewed the manuscript entitled “The emerging importance of cirsimaritin in type 2 diabetes treatment”, and it shows an interesting approach on the use of cirsimaritin in the treatment of type 2 diabetes in rats. The methodology employed is adequate, the idea of the manuscript is original and the subject is suitable for the International Journal of Molecular Sciences. However, some grammar and proofing issues need to be considered before possible publication in the IJMS.
R: Thank you very much for your positive feedback.
C1: Abstract, line 5: It is unclear whether two dosages of cirsimaritin were evaluated.
R1: Apology for this confusion, the second dose is for metformin. We corrected it.
C2: Introduction: Species names in the last paragraph of the Introduction must be italicized.
R2: Thank you for your comment. Corrected now.
C3: Results: There are captions for Figures 5 and 7 but the images were not provided in the manuscript.
R3: We apologize for this mistake. The figures are added now.
C4: Discussion: The discussion of the manuscript needs to be improved. The authors can better explain the importance of new medicines based on natural products. You can read on that in: https://doi.org/10.1007/s11011-017-0089-y and https://doi.org/10.1371/journal.pone.0255996
R4: Thank you very much for your comment. A paragraph is added now to the discussion to explain the importance of new medicines based on natural products.
C5: Material and methods: Can you provide the average weight of the rats before and after the T2D induction?
R5: Thank you for your comment. The average weight was added to the methodology section.
C6: Material and methods: It is necessary to explain how cirsimaritin was obtained and purified. Was it purchased from a commercial source (insert manufacturer) or extracted from a plant (insert methodology employed)?
R6: Thank you for your comment. Cirsimaritin used in this study is a pure compound, and the manufacturer is added now.
C7: Manuscript grammar must be carefully proofread.
R7: Thank you for your comment. The manuscript was gone through proofreading.
Reviewer 3 Report
This manuscript investigated the anti-diabetic effects of cirsimaritin in a high-fat diet and streptozotocin-(HFD/STZ)-induced rat model of type 2 diabetes mellitus (T2D). The topic and results are interesting and I think it can be accepted for publication in its current form.
Author Response
Dear editor and reviewers
We would like to thank you for the invested time and effort in carefully reviewing our manuscript.
C: This manuscript investigated the anti-diabetic effects of cirsimaritin in a high-fat diet and streptozotocin-(HFD/STZ)-induced rat model of type 2 diabetes mellitus (T2D). The topic and results are interesting and I think it can be accepted for publication in its current form.
R: Thank you very much for your positive feedback.
Reviewer 4 Report
This manuscript is quite interesting. The authors explained the benefits of one natural product, namely: cirsimaritin.
1. The authors should show the structure of an active compound to compare it with standard medicine.
2. Abstract: quite clear.
3. Introduction part: p. 2, Please print the scientific name of plants in italics in the third paragraph. However, please check all names of the plants as well.
3. Results part: Topic 2.1, 2.2, and 2.3 should write the name of cirsimaritin in capital letters (Cirsimaritin).
4. Did you forget to put Figure 5 and Figure 7 in the manuscript, by chance?
5. Discussion part; please rewrite the sentence in the first paragraph.
6. In the experiment, why did you treat cirsimaritin 50 mg/Kg and metformin 200 mg/kg with rats? Can you use it in the exact dose?
7. On page 9, You should briefly put the detail of reference 45.
Author Response
Dear editor and reviewers
We would like to thank you for the invested time and effort in carefully reviewing our manuscript. We are grateful for giving us the opportunity to revise our manuscript. Your comments were very useful and helped us in improving our manuscript. After careful consideration of the comments, the revision included many positive changes as suggested. Changes are highlighted using track changes.
C: This manuscript is quite interesting. The authors explained the benefits of one natural product, namely: cirsimaritin.
R: Thank you very much for your positive feedback.
C1: The authors should show the structure of an active compound to compare it with standard medicine.
R1: Thank you for your comment. The structure for the compound is added now as figure 1.
C2: Abstract: quite clear.
R2: Many thanks for your positive feedback.
C3: Introduction part: p. 2, Please print the scientific name of plants in italics in the third paragraph. However, please check all names of the plants as well.
R3: Thank you for your comments. The plants names are italicized.
C4: Results part: Topic 2.1, 2.2, and 2.3 should write the name of cirsimaritin in capital letters (Cirsimaritin).
R4: Thank you for your comment. It is written now in capital.
C5: Did you forget to put Figure 5 and Figure 7 in the manuscript, by chance?
R5: We apologize for this mistake. The figures are now added to the manuscript.
C6: Discussion part; please rewrite the sentence in the first paragraph.
R6: Thank you for your comment. The sentence is re-written now.
C7: In the experiment, why did you treat cirsimaritin 50 mg/Kg and metformin 200 mg/kg with rats? Can you use it in the exact dose?
R7: Thank you for your comment. Doses of cirismaritin and metformin were chosen based on previous studies which are added now to the methodology section.
C9: On page 9, You should briefly put the detail of reference 45.
R9: Thank you for your comment. Brief details of reference 45 is added now to the discussion.